# Fish Oil Enriched Intravenous Lipid Emulsions Reduce Triglyceride Levels in Non-Critically Ill Patients with TPN and Type 2 Diabetes. A Post-Hoc Analysis of the INSUPAR Study

**DOI:** 10.3390/nu12061566

**Published:** 2020-05-27

**Authors:** Jose Abuín-Fernández, María José Tapia-Guerrero, Rafael López-Urdiales, Sandra Herranz-Antolín, Jose Manuel García-Almeida, Katherine García-Malpartida, Mercedes Ferrer-Gómez, Emilia Cancer-Minchot, Luis Miguel Luengo-Pérez, Julia Álvarez-Hernández, Carmen Aragón Valera, Julia Ocón-Bretón, Álvaro García-Manzanares, Irene Bretón-Lesmes, Pilar Serrano-Aguayo, Natalia Pérez-Ferre, Juan José López-Gómez, Josefina Olivares-Alcolea, Carmen Arraiza-Irigoyen, Cristina Tejera-Pérez, Jorge Daniel Martínez-González, Ana Urioste-Fondo, Ángel Luis Abad-González, María José Molina-Puerta, Ana Zugasti-Murillo, Juan Parra-Barona, Irela López-Cobo, Gabriel Olveira

**Affiliations:** 1Unidad de Gestión Clínica de Endocrinología y Nutrición, Hospital Regional Universitario de Málaga—Instituto de Investigación Biomédica de Málaga (IBIMA), 29010 Málaga, Spain; jose.abuin.fdez@gmail.com (J.A.-F.); mjtapiague@gmail.com (M.J.T.-G.); 2Departmento de Medicina y Dermatología, Universidad de Málaga, 29071 Málaga, Spain; 3Servicio de Endocrinología y Nutrición, Hospital Universitari de Bellvitge, 08907 L’Hospitalet de Llobregat (Barcelona), Spain; rlurdiales@gmail.com; 4Servicio de Endocrinología y Nutrición, Hospital Universitario de Guadalajara, 19002 Guadalajara, Spain; herranzantolin@yahoo.es; 5Servicio de Endocrinología y Nutrición, Hospital Universitario Virgen de la Victoria, 29010 Málaga, Spain; jgarciaalmeida@yahoo.com; 6Servicio de Endocrinología y Nutrición, Hospital Universitario y Politécnico La Fe, 46026 Valencia, Spain; kathe.garciamalpartida@gmail.com; 7Servicio de Endocrinología y Nutrición, Hospital Clínico Universitario Virgen de la Arrixaca, 30120 Murcia, Spain; tofly16@gmail.com; 8Sección de Endocrinología y Nutrición, Hospital Universitario de Fuenlabrada, 28942 Madrid, Spain; emilia.cancer@salud.madrid.org; 9Servicio de Endocrinología y Nutrición, Hospital Universitario de Badajoz, 06080 Badajoz, Spain; luismiluengo@yahoo.es; 10Servicio de Endocrinología y Nutrición, Hospital Universitario Príncipe de Asturias, 28805 Madrid, Spain; julia.alvarez@movistar.es; 11Servicio de Endocrinología y Nutrición, Hospital Universitario Fundación Jiménez Díaz, 28040 Madrid, Spain; caragonva@fjd.es; 12Servicio de Endocrinología y Nutrición, Hospital Universitario Lozano Blesa, 50009 Zaragoza, Spain; mjocon@salud.aragon.es; 13Servicio de Endocrinología y Nutrición, Hospital General La Mancha Centro, 13600 Alcázar de San Juan, Spain; agmanzanares2010@gmail.com; 14Servicio de Endocrinología y Nutrición, Hospital Universitario Gregorio Marañón, 28007 Madrid, Spain; irene.breton@salud.madrid.org; 15Unidad de Endocrinología y Nutrición, Hospital Universitario Virgen del Rocío, 41013 Sevilla, Spain; piagua@gmail.com; 16Servicio de Endocrinología y Nutrición, Hospital Clínico San Carlos, 28040 Madrid, Spain; npferre@salud.madrid.org; 17Servicio de Endocrinología y Nutrición, Hospital Clínico Universitario de Valladolid, 47005 Valladolid, Spain; jjlopez161282@hotmail.com; 18Servicio de Endocrinología y Nutrición, Hospital Universitario Son Llatzer, 07198 Illes Balears, Spain; josefinaolivares@gmail.com; 19Servicio de Endocrinología y Nutrición, Complejo Hospitalario de Jaén, 23007 Jaén, Spain; mc.arraiza.sspa@juntadeandalucia.es; 20Servicio de Endocrinología y Nutrición, Complejo Hospitalario Universitario de Ferrol, 15045 A Coruña, Spain; cristinatejera.mui@gmail.com; 21Servicio de Endocrinología y Nutrición, Hospital Universitario Severo Ochoa, 28914 Leganés (Madrid), Spain; jdmartinglez@gmail.com; 22Servicio de Endocrinología y Nutrición, Complejo Asistencial Universitario de León, 24071 León, Spain; anaurifon@gmail.com; 23Unidad de Nutrición – Sección de Endocrinología, Hospital General Universitario de Alicante, 03010 Alicante, Spain; angeluis1024@gmail.com; 24Servicio de Endocrinología y Nutrición, Hospital Universitario Reina Sofía, 14004 Córdoba, Spain; cmmerinomjmolina@hotmail.com; 25Servicio de Endocrinología y Nutrición, Complejo Hospitalario de Navarra, 31008 Navarra, Spain; azugas@hotmail.com; 26Servicio de Endocrinología y Nutrición, Hospital de Mérida, 06800 Badajoz, Spain; juanparrabarona@gmail.com; 27Servicio de Endocrinología y Nutrición, Hospital de Sant Joan Despí Moisès Broggi, 08970 Barcelona, Spain; irela.lopez@sanitatintegral.org; 28CIBERDEM (CB07/08/0019), Instituto de Salud Carlos III, 28029 Madrid, Spain

**Keywords:** parenteral nutrition, type 2 diabetes mellitus, polyunsaturated fatty acids, omega 3, hospital

## Abstract

There are no studies that have specifically assessed the role of intravenous lipid emulsions (ILE) enriched with fish oil in people with diabetes receiving total parenteral nutrition (TPN). The objective of this study was to assess the metabolic control (glycemic and lipid) and in-hospital complications that occurred in non-critically ill inpatients with TPN and type 2 diabetes with regard to the use of fish oil emulsions compared with other ILEs. We performed a post-hoc analysis of the Insulin in Parenteral Nutrition (INSUPAR) trial that included patients who started with TPN for any cause and that would predictably continue with TPN for at least five days. The study included 161 patients who started with TPN for any cause. There were 80 patients (49.7%) on fish oil enriched ILEs and 81 patients (50.3%) on other ILEs. We found significant decreases in triglyceride levels in the fish oil group compared to the other patients. We did not find any differences in glucose metabolic control: mean capillary glucose, glycemic variability, and insulin dose, except in the number of mild hypoglycemic events that was significantly higher in the fish oil group. We did not observe any differences in other metabolic, liver or infectious complications, in-hospital length of stay or mortality.

## 1. Introduction

Although enteral nutrition is the first option in patients that require nutritional support, total parenteral nutrition is a more appropriate technique to reduce complications and mortality in malnourished patients or in those at risk of being malnourished and who cannot use their digestive tract [1].

Lipid emulsions containing fish oil are rich in n-3 polyunsaturated fatty acids (PUFAs) such as docosahexaenoic acid (DHA) and eicosapentaenoic acid (EPA), which exhibit anti-inflammatory, immunomodulatory, and antioxidative properties in preclinical models. In recent years the evidence to support the use of intravenous lipid emulsions (ILEs) enriched with fish oil has grown increasingly [2,3,4], showing that they could lower triglyceride concentrations, inflammatory markers, and liver function enzymes, and improve morbidity (risk of infection and sepsis and length of stay) and even mortality outcomes in hospitalized patients especially in post-surgical and oncology patients [5,6,7,8], when compared with ILEs based on soybean oil.

The European Society of Parenteral and Enteral Nutrition (ESPEN) Expert Group supports the use of olive oil and fish oil (FO) in nutrition support in surgical and non-surgical intensive care unit (ICU) patients, but considers that further research is required to provide a more robust evidence base [9].

It has been reported that n-3 PUFA status is inversely associated with type 2 diabetes and its oral supplementation could reduce insulin resistance, especially in women [10], and could prevent the onset of diabetes [11]. In animal models, intravenously infused n-3 PUFA preserves insulin signaling and glucose uptake compared to the infusion of n-6 PUFA [12].

In people with type 2 diabetes, fish oil supplementation lowers triglycerides but it does not seem to have any statistically significant effect on glycemic control [13].

Currently, there are no specific studies in people with diabetes receiving parenteral nutrition that have assessed the role of emulsions enriched with n-3 PUFA with regard to glycemic and lipid control, nor regarding possible complications.

In the INSUPAR trial, previously published by our group, we assessed two insulin regimens in adult inpatients with type 2 diabetes in a non-critical setting with indication for total parenteral nutrition (TPN; subcutaneously administered glargine insulin vs. regular insulin inside the TPN bag) [14]. Of the 161 patients assessed, 80 (49.7%) received TPN with fish oil enriched emulsions and 81 other ILEs. As this was a clinical trial, we could assess the effect of n-3 PUFA enriched ILEs with regard to other emulsions. Our hypothesis was that enriched n-3 PUFA ILEs could modulate glycemic control and reduce hypertriglyceridemia and complications in these patients.

Therefore, the objective of this post-hoc analysis of the INSUPAR trial was to assess the metabolic control and in-hospital complications that occurred in non-critically ill inpatients with TPN and type 2 diabetes with regard to the use of enriched n-3 PUFA emulsions compared with the other ILEs.

## 2. Research Design and Methods

The results were extracted from the INSUPAR trial [14] (complete details available in the main article). The study included >18 years non-critically ill type 2 diabetes in-hospital patients who started with TPN (considering that it provides more than 70% of the estimated total energy expenditure) for any cause and that would predictably continue with TPN for at least five days. All TPN were infused by central line (ClinicalTrials.gov NCT02706119 and EudraCT 2015-003954-42).

Patients were considered to have diabetes as assessed according to the international criteria [15]. Blood glucose levels were obtained from capillary and the same glucose meter was provided (Freestyle Optium; Abbott Diabetes Care Inc, Witney, Oxon, UK) to every center. Measurements were made four times a day. Optimum blood glucose levels were between 100 and 140 mg/dL. Blood glucose measurements were performed until the patient discontinued TPN or up to 15 days at most. We continued to monitor capillary glucose on days 1 and 2 after TPN was stopped.

Hypoglycemia was defined as blood glucose ≤70 mg/dL [16]. Glycemic variability (GV) was measured by standard deviation and variation coefficient of capillary glucose.

The following baseline data were recorded: demographic variables, treatment modality, diagnosis on admission, prior comorbidity (Charlson Comorbidity Index [17]), anthropometric data (body mass index (BMI)), nutritional assessment by subjective global assessment (SGA), type 2 diabetes related parameters, TPN characteristics, and concomitant prescription of drugs that could induce hyperglycemia.

### 2.1. Analytical Assessment during TPN Infusion 

Analytical follow-up was made according to common clinical practice in each center, but at least a blood count, coagulation, and biochemistry were made on days 1 and 5 since the beginning of TPN, and the day before stopping it. A fasting blood sample was drawn to measure: the glycated hemoglobin (following the international recommendations for standardization of the HbA1c measurement [18]), plasma blood glucose, C-reactive protein (CRP), triglycerides, cholesterol, creatinine, urea, electrolytes, and blood liver function at the laboratories of each hospital (with an autoanalyzer).

### 2.2. Metabolic and Liver Complications

Urea increase was considered if the previous value was normal and then increased above 80 mg/dL after the beginning of TPN. Creatinine increase was considered if the previous value was normal and then increased above 1.3 mg/dL after the beginning of TPN. Any analytical value above or below the following normal ranges was considered a complication: hypertriglyceridemia (>400 mg/dL) [4], hypernatremia (>150 mEq/L), hyponatremia (<135 mEq/L), hypokalemia (<3 mEq/L), hypomagnesemia (<1.2 mg/dL), hypophosphatemia (<2 mg/dL), hyperchloremia (>120 mEq/L), and hypocalcemia (with corrected Calcium for Albumin; <8 mg/dL).

Liver function was considered altered when liver enzymes increased above twice the upper normal limit in at least two of its parameters (being previously normal): Aspartate Transaminase (AST), Alanine Transaminase (ALT), Gamma-Glutamyltransferase (GGT), and Alkaline Phosphatase after the beginning of TPN.

Other complications were evaluated from the beginning of TPN to the discharge of the patients, not only during TPN infusion:Infectious non-catheter and catheter related bloodstream infections; they were identified as an elevated white blood cell count in addition to one or more of the following: positive blood cultures, chest x-ray suggestive of pneumonia, positive urine culture, postoperative wound infection, and use of antibiotics.Length of stay.In-hospital mortality.

### 2.3. Statistical Analysis

Data analysis was performed using SPSS version 22.0 (Armonk, NY, USA) [19]. The Kolmogorov–Smirnov test was used to assess whether the variables were normally distributed or not. The hypothesis contrast between proportions was done using the *χ*^2^ test with Fisher’s exact test, when necessary. Hypothesis contrast for continuous variables between groups used the *t*-test for variables that followed a normal distribution, and a non-parametric test (Mann–Whitney) for variables that did not conform to normal. In the case that significant differences between groups were found in any of the baseline characteristics, we used a general lineal model adjusted by these covariates (sex and TPN duration).

Variables tested repeatedly over time (glucose, CRP, triglycerides, cholesterol, and blood liver function) were also analyzed using repeated measures multiple analysis of variance according to time and group of ILEs.

For all the calculations, significance was set at *p* < 0.05 for two tails. 

## 3. Results

### Sample

A total of 161 patients were included in the INSUPAR study (80 in the Regular Insulin group and 81 in the Glargine group). Of them, 80 patients (49.7%) were on ILEs enriched with n-3 PUFA and 81 patients (50.3%) on other ILEs, where 49 (27.5%) received a mixture of medium and long chain triglycerides (MCT/LCT; Lipofundina MCT/LCT 20%; B. Braun Medical, Rubí, Barcelona, Spain), 29 (16.3%) based on olive oil (Clinoleic 20%; Baxter, Ribarroja del Turia, Valencia, Spain), and three (1.7%) based on pure soybean oil (Intralipid; Fresenius Kabi España, Barcelona, Spain. Of the 80 patients with n-3 PUFA, 49 (61.3%) of them were with Smoflipid (Fresenius Kabi AB, Uppsala, Sweden) and 31 (38.7%) of them were on Lipoplus (B. Braun Melsungen AG, Melsungen, Germany). The mean daily n-3 PUFA intake was 4.34 ± 1.19 g and 0.067 ± 0.0201 g/kg.

We did not find any differences in the baseline characteristics between both groups other than a higher percentage of men and higher duration of TPN in the n-3 PUFA group (Table 1).

We did not find any differences in glucose metabolic control (mean capillary glucose, glycemic variability and insulin dose) except for the number of mild hypoglycemic events being significantly higher in the n-3 PUFA group (Table 2). We did not observe any differences in other metabolic, liver or infectious complications, in-hospital length of stay, or mortality. We observed a significant decrease in triglyceride levels in the n-3 PUFA group on day 5 and the last day of TPN compared to day 1 and significant differences between groups (Table 3 and Figure 1).

## 4. Discussion

In this study, we observed that in non-critically ill patients with type 2 diabetes mellitus (T2DM) who received TPN with n-3 PUFA enriched ILEs (compared to other ILEs), triglyceride levels were significantly reduced and presented a higher number of mild hypoglycemia events. On the other hand, we did not find any differences in other parameters of glycemic or lipid metabolic control, liver or infectious complications, hospital length of stay or mortality.

Lipid emulsions are used in parenteral nutrition (PN) with the objective of supplying an energy-dense source of calories, reducing the glycemic load, supplying essential fatty acids, and lowering osmolarity [8]. The first generation of lipid emulsions was based on soybean oil and contained high concentrations of n-6 PUFA that could promote inflammation [3]. Fish oil contains ω-3 PUFAs (docosahexaenoic acid [DHA] and eicosapentaenoic acid [EPA]) that incorporate into cell membranes influencing various transcription factors modifying the expression of genes involved in many biological processes which include metabolism, immune function, and inflammation [3]. ILEs based on fish oil have been shown to have anti-inflammatory and immunomodulatory effects [5].

### 4.1. Lipid Control and Liver Enzymes

Oral fish oil supplementation has been shown to produce a clinically significant, dose-dependent reduction in fasting blood triglycerides and normalize serum lipid concentrations, including high-density lipoproteins (HDL) and low-density lipoproteins (LDL), in patients with diabetes mellitus [2]. In patients with PN it is not uncommon to experience increases in serum triglyceride concentrations and hypertriglyceridemia correlated with hepatic steatosis, which can contribute to liver damage. In this regard, ILEs based on fish oil may be protective against rapid increases in serum triglycerides, compared with other formulas such as those based on MCT/LCT, although the evidence is not conclusive [5,20]. In our study we did not observe any differences in the percentage of patients that developed hypertriglyceridemia (levels above 400 mg/dL) which was, in any case, low in both groups (5% in the n-3 PUFA group and 7.4% in the other group); nonetheless, we observed significant reductions of levels of serum triglycerides in the n-3 PUFA group.

Lipid composition of TPN could also play a significant role in liver enzyme alteration associated with PN, and n-3 PUFA ILEs have been shown to minimize this disturbance in hospitalized adult patients by reducing liver complications [21,22]. The administration of high doses of intravenous lipids, that are high in n-6 PUFAs and phytosterols (like the ones based on soybean oils), can contribute to the development of parenteral nutrition-associated liver disease [23]. ILEs based exclusively on fish oil reduce plasmatic phytosterol levels and are associated with an improvement in liver profile [24,25]. In our study, regarding analytical data, we did not find any differences in liver profile, neither in the percentage of patients with an increase in liver enzymes nor in the number of patients with liver complications. This could be because the use of ILEs based on pure soybean oil was very low (only three patients, 1.7% of the sample) in our sample. Therefore, even the rest of the formulas used in our study (MCT/LCT, olive oil, and fish oil emulsions) contain variable quantities of n-6 PUFA (but always less than pure soybean oils), and these ILEs could have advantages due to the reduced accumulation of phytosterols and because of the mitigation of the proinflammatory effect of n-6 PUFA, as well as potentiate positive effects of other lipid sources [26].

### 4.2. Complications

Several studies have also shown a significant decrease in hospital stay, especially in ICU surgical patients [3,5,7,9,27] and in infectious complications [2,28,29] when using ILEs based on fish oil compared mainly with ILEs based on pure soybean oil or MCT/LCT emulsions. A reduction of infectious complications is seen with the doses that are usually applied in clinical practice [3] and is supported by the observations that using these fish oil containing ILEs can decrease the blood concentrations or ex-vivo production of proinflammatory eicosanoids and cytokines [3,5]. In our sample we administered similar doses to those used in other studies [3], but we did not find any differences in these parameters and also C reactive protein levels decreased similarly in both groups.

A possible cause of the lack of differences compared to other studies could be a limited sample but also due to the fact that we were comparing enriched n-3 PUFA emulsions vs. other ILEs that are not based on n-6 PUFA as the only source and thus these already have an improved composition (50% MCT and olive oil ILEs) [3,8].

### 4.3. Glycemic Control

With regard to glycemic control, it was also similar when comparing both groups with regard to mean capillary blood glucose, insulin dose, and glycemic variability; however, we observed a small but significantly higher number of mild hypoglycemia events in the n-3 PUFA group. This could be explained due to a higher insulin sensitivity in patients using n-3 PUFA [10]. Besides this, patients with n-3 PUFA ILEs had a non-statistically significant tendency to present a higher proportion of patients with T2DM with end-organ damage and significantly longer TPN duration, both variables that have been associated with a higher risk of hypoglycemia in T2DM patients [30]. On the other hand, they did not have greater glycemic variability differences in the percentage of patients with glargine or regular insulin [30].

### 4.4. Limitations and Conclusions

The main advantages of our study are that it is a prospective multi-center study, with a considerable sample (compared to other studies), and very homogeneous (all of them previously diagnosed with T2DM) and it involves a follow-up of the patients from admission to discharge, not only during TPN infusion. However, the current study is not free from limitations as it is a post-hoc analysis, we only focus on non-critically ill patients with type 2 diabetes mellitus; therefore, we cannot apply these conclusions to other groups of patients. On the other hand, there could be differences regarding the cause of the admission (surgical, oncohematological, or medical) that we have not assessed.

In conclusion, although the results come from a post-hoc analysis of our previous study, we found that in non-critically ill patients with T2DM who received TPN with lipid emulsions enriched with n-3 PUFA, triglyceride levels were significantly reduced and presented a higher number of mild hypoglycemia events without differences in glycemic control and in-hospital complications compared to other lipid emulsions. Future prospective research is needed to examine the effect of n-3 PUFA on patients with diabetes receiving PN.

## Figures and Tables

**Figure 1 nutrients-12-01566-f001:**
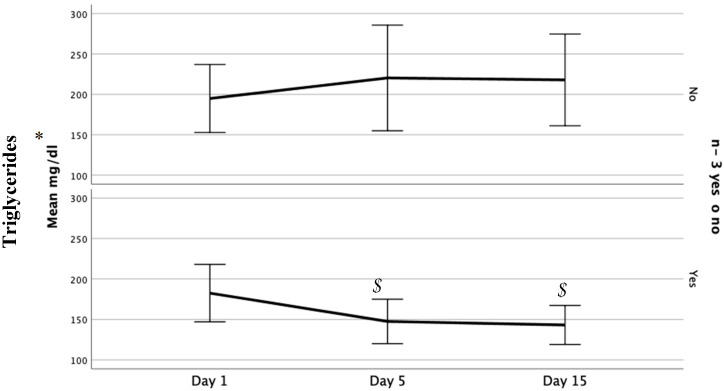
Evolution of Triglyceride levels by group. * Differences between groups by repeated measures multiple analysis of variance; $ *p* < 0.01 in the n-3 polyunsaturated fatty acids (PUFA) group between day 1 and 5 and between day 1 and the last day.

**Table 1 nutrients-12-01566-t001:** Baseline characteristics.

Variable	Other ILEs *n* = 81 (50.3%)	n-3 PUFA *n* = 80 (49.7%)	*p*-Value
Gender			
Men, *n* (%)	47 (58.0%)	63 (78.8%)	**0.004**
Women, *n* (%)	34 (42.0%)	17 (21.2%)	
Age (years)	71.6 ± 10.3	70.4 ± 9.5	0.451
Group of treatment			
Regular insulin, *n* (%)	40 (49.4%)	40 (50.0%)	0.938
Glargine insulin, *n* (%)	41 (50.6%)	40 (50.0%)	
Subjective global assessment			
Well nourished, *n* (%)	28 (34.6%)	24 (30.0%)	0.549
Moderate malnutrition, *n* (%)	32 (39.5%)	29 (36.3%)	
Severe malnutrition, *n* (%)	21 (25.9%)	27 (33.7%)	
Blood test parameters			
Creatinine (mg/dL)	0.84 ± 0.24	0.77 ± 0.25	0.089
Albumin (mg/dL)	2.57 ± 0.52	2.47 ± 0.57	0.219
C reactive protein (mg/dL)	112.3 ± 103.6	90.2 ± 90.9	0.169
Reason for admission			
Surgical, *n* (%)	41 (50.6%)	43 (53.8%)	0.242
Oncohematological, *n* (%)	22 (27.2%)	27 (33.7%)	
Medical, *n* (%)	18 (22.2%)	10 (12.5%)	
Charlson comorbidity index	6.5 ± 2.9	7.1 ± 2.9	0.188
Type 2 diabetes mellitus			
Duration of type 2 diabetes mellitus (years)	11.0 ± 8.5	11.3 ± 7.3	0.840
Diabetes with end-organ damage, *n* (%)	10 (12.3%)	15 (18.8%)	0.262
Patients with insulin prior to the admission, *n* (%)	20 (24.7%)	26 (32.5%)	0.230
Insulin units prior to the admission (IU/kg/day)	0.36 ± 0.23	0.55 ± 0.34	0.075
HbA1c (%)	6.60 ± 1.0	6.6 ± 1.2	0.912
Plasma glucose (mg/dL)	190.3 ± 64.3	184.3 ± 81.5	0.604
Real weight (kg)	71.16 ± 17.33	74.14 ± 15.73	0.255
BMI (kg/m^2^)	27.14 ± 5.50	27.22 ± 5.88	0.919
<18.5 kg/m^2^	1 (1.2%)	3 (3.7%)	0.724
18.5–25.0 kg/m^2^	30 (37.0%)	27 (33.8%)	
25.0–30.0 kg/m^2^	32 (39.5%)	30 (37.5%)	
>30.0 kg/m^2^	18 (22.2%)	20 (25.0%)	
Total energy expenditure (kcal)	1599.0 ± 248.7	1636.5 ± 210.0	0.303
Any drug induces hyperglycemia, *n* (%)	17 (21.0%)	20 (25.0%)	0.545
TPN duration (days)	8.6 ± 4.3	11.6 ± 8.8	**0.007**

BMI, body mass index; HbA1c, glycated hemoglobin; ILEs, intravenous lipid emulsions; PUFA, polyunsaturated fatty acids; TPN, total parenteral nutrition. Bold *p*-values indicate statistical significance.

**Table 2 nutrients-12-01566-t002:** Glycemic-related variables.

	Other ILEs *n* = 81 (50.3%)	n-3 PUFA *n* = 80 (49.7%)	*p*-Value *
Insulin			
Mean total daily insulin (IU/kg)	45.7 ± 26.1	47.5 ± 26.8	0.662
Mean total daily insulin/10 g of carbohydrates in TPN (IU)	2.5 ± 1.2	2.5 ± 1.3	0.894
Mean capillary glucose			
08:00 h (mg/dL)	169.1 ± 41.3	163.6 ± 40.4	0.398
13:00 h (mg/dL)	176 ± 42.6	172.1 ± 42.3	0.560
20:00 h (mg/dL)	174.1 ± 44.5	163 ± 38.6	0.100
00:00 h (mg/dL)	164.7 ± 44.6	159.1 ± 42.3	0.446
During TPN (mg/dL)	172.4 ± 40.9	165.4 ± 38.6	0.269
Mean post-TPN capillary blood glucose 48 h (mg/dL)	156 ± 49.1	145.5 ± 40.5	0.206
Hypoglycemic variables			
Number of capillary glucose ≤ 70 mg/dL, *n* (%)	0.18 ± 0.61	0.55 ± 1.17	**0.012**
Number of capillary glucose < 54 mg/dL, *n* (%)	0.04 ± 0.19	0.06 ± 0.24	0.471
Capillary glucose variability			
Standard deviation of capillary glucose (mg/dL)	41.3 ± 18.7	42.6 ± 16.5	0.634
Variation coefficient of capillary glucose (%)	24 ± 9.2	25.9 ± 9.2	0.187

* Adjusted for gender and duration of TPN. Bold *p*-values indicate statistical significance.

**Table 3 nutrients-12-01566-t003:** Complications.

		Other ILEs *n* = 81 (50.3%)		n-3 PUFA *n* = 80 (49.7%)	*p*-Value *	
Metabolic and liver complications						
Hypertriglyceridemia, *n* (%)		6 (7.4%)		4 (5.0%)	0.527	
Hypernatremia, *n* (%)		1 (1.2%)		5 (6.3%)	0.093	
Hyponatremia, *n* (%)		2 (2.5%)		7 (8.8)	0.083	
Hypokalemia, *n* (%)		7 (8.6%)		8 (10.0%)	0.767	
Hypophosphatemia, *n* (%)		8 (9.9%)		11 (13.8%)	0.446	
Hypocalcemia (with corrected calcium), *n* (%)		2 (2.5%)		2 (2.5%)	0.980	
Increased creatinine, *n* (%)		5 (6.2%)		4 (5.0%)	0.732	
Increased urea, *n* (%)		5 (6.2%)		7 (8.8%)	0.533	
Liver complications, *n* (%)		5 (6.2%)		7 (8.8%)	0.534	
Infectious and other complications						
Central line-associated bloodstream infections, *n* (%)		6 (7.4%)		11 (13.8%)	0.190	
Sepsis, *n* (%)		4 (4.9%)		6 (7.5%)	0.501	
Pneumonia, *n* (%)		4 (4.9%)		2 (2.5%)	0.414	
Surgical site infection, *n* (%)		5 (6.2%)		8 (10.0%)	0.360	
Urinary tract infection, *n* (%)		3 (3.7%)		2 (2.5%)	0.660	
Mortality, *n* (%)		10 (12.3%)		14 (17.5%)	0.359	
Length of hospital stay (days)		28.7 ± 20.0		32.2 ± 27.5	0.367	
Blood test results	Days		Days			Analysis of variance
Triglycerides (mg/dL) ^$^	1 (*n* = 79)	194.8 ± 86.8	1 (*n* = 79)	182.5 ± 93.9	0.663	**0.028**
	5 (*n* = 70)	220.3 ± 134.8	5 (*n* = 68)	147.5 ± 72.6 ^&^	0.052	
	Last (*n* = 19)	217.9 ± 117.1	Last (*n* = 28)	143.1 ± 63.7 ^&^	**0.024**	
Total cholesterol (mg/dL)	1 (*n* = 79)	115.9 ± 48.3	1 (*n* = 79)	117.7 ± 51.2	0.825	0.220
	5 (*n* = 72)	123.4 ± 37.1	5 (*n* = 65)	114.7 ± 33.8	0.156	
	Last (*n* = 19)	135.2 ± 44.9	Last (*n* = 26)	119.9 ± 35.3	0.206	
HDL cholesterol (mg/dL)	1 (*n* = 50)	23.6 ± 12.2	1 (*n* = 51)	23.1 ± 11.4	0.831	0.401
	5 (*n* = 45)	21.9 ± 8.9	5 (*n* = 43)	20.1 ± 9.4	0.366	
	Last (*n* = 7)	18.9 ± 6.5	Last (*n* = 19)	19.2 ± 8.6	0.934	
LDL cholesterol (mg/dL)	1 (*n* = 49)	50.1 ± 26.6	1 (*n* = 47)	58.6 ± 42.4	0.246	0.770
	5 (*n* = 44)	61.8 ± 30.9	5 (*n* = 40)	58.3 ± 28.7	0.586	
	Last (*n* = 6)	73.2 ± 40.4	Last (*n* = 18)	69.4 ± 32.1	0.817	
Aspartate transaminase (U/L)	1 (*n* = 78)	29.1 ± 33.1	1 (*n* = 78)	29.7 ± 36.5	0.936	0.797
	5 (*n* = 67)	35.0 ± 37.2	5 (*n* = 59)	35.9 ± 38.0	0.897	
	Last (*n* = 20)	32.0 ± 27.4	Last (*n* = 32)	31.3 ± 35.9	0.939	
Alanine aminotransferase (U/L)	1 (*n* = 78)	30.6 ± 32.9	1 (*n* = 78)	29.9 ± 33.4	0.904	0.319
	5 (*n* = 72)	38.8 ± 49.2	5 (*n* = 68)	32.4 ± 30.6	0.363	
	Last (*n* = 21)	35.2 ± 44.9	Last (N = 34)	30.4 ± 26.7	0.624	
Gamma glutamyl transferase (U/L)	1 (*n* = 78)	98.2 ± 131.4	1 (*n* = 78)	151.5 ± 183.3	0.057	0.303
	5 (*n* = 64)	177.2 ± 155.7	5 (*n* = 58)	199.4 ± 185.2	0.474	
	Last (*n* = 19)	171.4 ± 155.0	Last (*n* = 28)	211.6 ± 205.4	0.472	
Alcaline phosphatase (U/L)	1 (*n* = 78)	115.8 ± 111.6	1 (*n* = 78)	139.8 ± 197.9	0.339	0.549
	5 (*n* = 68)	151.3 ± 125.3	5 (*n* = 66)	154.2 ± 186.9	0.918	
	Last (*n* = 20)	121.0 ± 60.0	Last (*n* = 32)	166.0 ± 160.0	0.234	
C reactive protein (mg/L)	1 (*n* = 71)	112.3 ± 103.6	1 (*n* = 77)	90.2 ± 90.9	0.169	0.320
	5 (*n* = 66)	68.7 ± 77.9	5 (*n* = 68)	65.3 ± 79.6	0.805	
	Last (*n* = 15)	58.4 ± 81.4	Last (*n* = 29)	56.2 ± 48.6	0.913	

HDL, high density lipoprotein; LDL, low density lipoprotein; * *p*-value between groups of ILEs adjusted for gender and TPN duration. ^$^ Differences between groups by repeated measures multiple analysis of variance according to time (days 1, 5, and 15) and group of ILEs. ^&^
*p* < 0.01 in the n-3 PUFA group between day 1 and 5 and between day 1 and the last day. Bold *p*-values indicate statistical significance.

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
