# Peer review of "Fish Oil Enriched Intravenous Lipid Emulsions Reduce Triglyceride Levels in Non-Critically Ill Patients with TPN and Type 2 Diabetes. A Post-Hoc Analysis of the INSUPAR Study"

_nutrients, 2020, doi:10.3390/nu12061566_

Round 1

Reviewer 1 Report

The authors provide a fine post-hoc analysis concerning the effect of n-3 PUFA in TPN on triglycerides, when administered in hospitalized T2DM patients. I have some minor comments:

  • The Discussion should mention more on the fact that patients were hospitalized for a variety of reasons (surgical, oncohematological and medical), which could have influenced the results (e.g. table 1 mentions much more patients hospitalized for medical reasons in the "other ILE" group, albeit not statistically significant).
  • Table 1 also mentions a "Subjective Global Assessment" concerning nutritional status, which should be explained in the Discussion (what/which measurements?), and mentioned as a confounding factor.
  • Given the effect on triglyderides in T2DM patients, is there any effect to be seen on the prevalence of acute pancreatitis (if these data were collected)?

Author Response

Dear Editors,

Thank you for giving us the opportunity to improve our article "Fish oil enriched intravenous lipid emulsions reduce triglyceride levels in non-critically ill patients with TPN and type 2 diabetes. A post-hoc analysis of the INSUPAR study.”, based on the comments of the reviewers and editors.

The various suggestions have been incorporated into the new version wherever applicable. Please find below our responses and the action taken to all the various suggestions and comments.

Once again, we very much appreciate all the work of the reviewers. Please thank them very specially on our behalf.

Yours sincerely,

Dr. Gabriel Olveira

________________________________________________________________

The authors provide a fine post-hoc analysis concerning the effect of n-3 PUFA in TPN on triglycerides, when administered in hospitalized T2DM patients. I have some minor comments:

The Discussion should mention more on the fact that patients were hospitalized for a variety of reasons (surgical, oncohematological and medical), which could have influenced the results (e.g. table 1 mentions much more patients hospitalized for medical reasons in the "other ILE" group, albeit not statistically significant).

Thank you very much for the appreciation. We have added a sentence to clarify this fact in the Discussion (Line 260-262).

Table 1 also mentions a "Subjective Global Assessment" concerning nutritional status, which should be explained in the Discussion (what/which measurements?), and mentioned as a confounding factor.

Thank you very much for the question. SGA is a well validated tool for assessment of nutritional status developed by Detsky et al. We have not mentioned it in the Discussion as we found no differences between groups.

Detsky AS, McLaughlin JR, Baker JP, et al. What is subjective global assessment of nutritional status? JPEN J Parenter Enteral Nutr 1987; 11:8–13.

Given the effect on triglycerides in T2DM patients, is there any effect to be seen on the prevalence of acute pancreatitis (if these data were collected)?

Thank you very much for the question. There were 5 patients that were with TPN because of severe acute pancreatitis but hypertriglyceridemia was not the cause and no patient developed pancreatitis during the admission. We decided not to include further analysis in our study because Triglycerides were similar in both groups on the admission and we did not have enough sample to do it.

Reviewer 2 Report

The aim of this trial was to assess the metabolic control and in-hospital complications occuring in hospitalized patients with type 2 diabetes mellitus receiving TPN enriched with n-2 PUFA or TPN with other ILE.

The article is well written and clearly presented.

Minor comments to the following:

line 46: there seems to be a typing error, a large Space between "n-3 PUFA" and "with"

line 83: there seems to be a typing error, a large Space between "glucose," and "C-reactive protein"

Figure 1: the symbols used in the figure "&" is uncommon. The name of the Y-axis should include "triglycerides". In the legend: The sentence does not make sense. Symbols should be presented on a line by itself. Hence, there should there be a full stop after "variance" and prior to "&"?.

Line 194: are you referring to complications in this section?

Line 195: inaccurate use of "," after "but".

Line 199: use ";" prior to "however" instead of ","

Line 209: there seems to be a typing error, a large Space between "from" and "admission"

Line 211: what do you meen with "previously designed one"? This sentence is very lo; hence, difficult to read. Please consider revision

Line 213-217 conclusion: i recommend the conclusion presents the non-significant differences in glycemic control and inhospital complications. These are important findings, although no difference was found and needs be presented in the conclusion.

Author Response

Dear Editors,

Thank you for giving us the opportunity to improve our article "Fish oil enriched intravenous lipid emulsions reduce triglyceride levels in non-critically ill patients with TPN and type 2 diabetes. A post-hoc analysis of the INSUPAR study.”, based on the comments of the reviewers and editors.

The various suggestions have been incorporated into the new version wherever applicable. Please find below our responses and the action taken to all the various suggestions and comments.

Once again, we very much appreciate all the work of the reviewers. Please thank them very specially on our behalf.

Yours sincerely,

Dr. Gabriel Olveira

________________________________________________________________

The aim of this trial was to assess the metabolic control and in-hospital complications occurring in hospitalized patients with type 2 diabetes mellitus receiving TPN enriched with n-2 PUFA or TPN with other ILE.

The article is well written and clearly presented.

Minor comments to the following:

line 46: there seems to be a typing error, a large Space between "n-3 PUFA" and "with"

Thank you for the appreciation, we have corrected the mistake.

line 83: there seems to be a typing error, a large Space between "glucose," and "C-reactive protein"

Thank you for the appreciation, we have corrected the mistake.

Figure 1: the symbols used in the figure "&" is uncommon. The name of the Y-axis should include "triglycerides". In the legend: The sentence does not make sense. Symbols should be presented on a line by itself. Hence, there should there be a full stop after "variance" and prior to "&"?.

Thank you for the appreciation, we have added "Triglycerides" in the Y axis and changed "&" for "$".

Line 194: are you referring to complications in this section?

Thank you for the question. Indeed, in this part of the Discussion we are referring to complications. We have divided the Discussion in section with subtitles to clarify (Lines 205, 231, 244 and 254).

Line 195: inaccurate use of "," after "but".

Thank you for the appreciation, we have corrected the mistake.

Line 199: use ";" prior to "however" instead of ","

Thank you for the appreciation, we have corrected the mistake.

Line 209: there seems to be a typing error, a large Space between "from" and "admission"

Thank you for the appreciation, we have corrected the mistake.

Line 211: what do you mean with "previously designed one"? This sentence is very lo; hence, difficult to read. Please consider revision

Thank you for the question. We refer to the INSUPAR trial, from which we gather data. We have removed part of the sentence to clarify.

Line 213-217 conclusion: I recommend the conclusion presents the non-significant differences in glycemic control and in-hospital complications. These are important findings, although no difference was found and needs be presented in the conclusion.

Thank you for the appreciation, we have added this information in Conclusions (Line 266).